# Cellular Senescence in Hepatocellular Carcinoma: The Passenger or the Driver?

**DOI:** 10.3390/cells12010132

**Published:** 2022-12-29

**Authors:** Xiurong Cai, Adrien Guillot, Hanyang Liu

**Affiliations:** 1Department of Hematology, Oncology and Tumor Immunology (CVK), Charité Universitätsmedizin Berlin, 13353 Berlin, Germany; 2Department of Hepatology and Gastroenterology (CVK), Charité Universitätsmedizin Berlin, 13353 Berlin, Germany; 3Center of Gastrointestinal Diseases, The Affiliated Changzhou Second People’s Hospital of Nanjing Medical University, Nanjing 213000, China

**Keywords:** liver cancer, cell-cycle arrest, senescence-targeting therapy, senescence-associated secretory phenotype

## Abstract

With the high morbidity and mortality, hepatocellular carcinoma (HCC) represents a major yet growing burden for our global community. The relapse-prone nature and drug resistance of HCC are regarded as the consequence of varying intracellular processes and extracellular interplay, which actively participate in tumor microenvironment remodeling. Amongst them, cellular senescence is regarded as a fail-safe program, leading to double-sword effects of both cell growth inhibition and tissue repair promotion. Particularly, cellular senescence serves a pivotal role in the progression of chronic inflammatory liver diseases, ultimately leading to carcinogenesis. Given the current challenges in improving the clinical management and outcome of HCC, senescence may exert striking potential in affecting anti-cancer strategies. In recent years, an increasing number of studies have emerged to investigate senescence-associated hepatocarcinogenesis and its derived therapies. In this review, we intend to provide an up-to-date understanding of liver cell senescence and its impacts on treatment modalities of HCC.

## 1. Introduction

Hepatocellular carcinoma (HCC), the most frequent type of primary liver malignancy, is a big threat to individuals and a huge challenge to health care systems worldwide [1]. HCC is still the third leading cause of cancer-related death globally, with a dismal prognosis of only 18% of the 5-year survival rate despite novel next-line options [2]. Although the risk factors for HCC are very diverse, sustained hepatic injury, fibrosis/cirrhosis and inflammation are commonly regarded as drivers for the initiation and progression of HCC [3,4,5]. Conventional treatments for HCC include surgical resection, liver transplantation, percutaneous ablation, intra-arterial or systemic chemotherapy, immune checkpoint blockade, as well as targeted therapies, which include kinase inhibitors and anti-angiogenesis reagents [6,7]. Although the targeted therapies listed above are suggested as promising cures for patients with advanced HCC, clinical benefits have only been revealed in a subgroup of patients with a modest survival extension in most instances [8]. Various molecular mechanisms and corresponding strategies have been proposed to understand and overcome treatment resistance in HCC. Particularly, a typical fail-safe process, cellular senescence, has been increasingly acknowledged as a central modulator in HCC during disease development and as a component of the treatment response.

Cellular senescence is initially described as a steady cell-cycle arrest, which is triggered by various stimuli, including continuous telomere shortening (‘replicative senescence’) [9], oncogene activation [10], oxidative damage, irradiation- and chemotherapy-induced DNA damage, which are termed ‘stress-induced senescence’ [11]. These stimuli trigger DNA damage response (DDR), leading to the activation of the p53 and the ERK/ETS1/2 pathways, which ultimately induce the expression of *p21^CIP1^* (also known as CDKN1A) and p16^INK4a^ (also known as CDKN2A), respectively [12,13]. With the overexpression of these two molecules as the blockers of cell-cycle progression, cells fail to enter the S phase from the G1 phase. Furthermore, unsolvable DDR activates the retinoblastoma (Rb) and p53 pathways, and promotes the formation of promyelocytic leukemia nuclear bodies, which ultimately leads to senescence-associated heterochromatin foci (SAHF) through the ASF1A and HIRA chaperones [14,15] (Figure 1). Senescence and apoptosis are equally important as intrinsic cellular programs that terminate successive expansion of the deleterious mutation-harboring cells, thereby serving as a fail-safe to prevent precancerous neoplasms and malignancy [16]. Senescent cells, unlike apoptotic cells, are still viable, metabolically active and capable of secreting multiple types of soluble factors, known as the senescence-associated secretory phenotype (SASP) [17]. Representing a source of matrix-modifying enzymes, growth factors, chemokines and inflammatory cytokines, SASP allows senescence to act as a double-edged sword, exerting distinct functions depending on its temporal-spatial regulation feature and the state of SASP-recipient cells [18]. Senescent cells can establish crosstalk with their neighboring cells through the dynamic changes in surface proteins and/or antigens presented by major histocompatibility complex (MHC) on senescent cells [19] and secreting biologically active materials via microvesicles and exosomes [20]. Through intricate cell–cell communication, senescent cells create either a pro-inflammatory or anti-inflammatory microenvironment that dynamically influences tissue homeostasis and disease progression at different stages including hepatitis, hepatic cirrhosis and HCC [21,22].

Based on current knowledge, we intent to summarize varying functions of cellular senescence and implications of senescence-targeting therapies (senescence-induced treatments and senescence-eliminating treatments including senolytics, senomorphics and immunotherapy) in HCC treatment (Figure 2). Particularly in this review, we will focus on how cellular senescence drives the interaction of parenchymal and non-parenchymal cells in hepatocarcinogenesis.

## 2. Cellular Senescence in Hepatocarcinogenesis

The liver serves a central role in the metabolic system, dominating the metabolism of nutrients, toxins and drugs. Consequently, both high workload and infectious risks make the liver prone to injury. Similarly, injured hepatic cells are terminated by cell death processes. Meanwhile, the liver possesses great regenerative abilities, which can either rescue impaired cells or generate substituting cells. Nonetheless, self-regeneration is not a cure for liver diseases. Continuous fibrosis and inflammation eventually lead to liver failure or malignant progression [23]. Intriguingly, senescence may act not only as an age-related biological process, but also as a concomitant mechanism of cell fate alterations [21,24]. In this section, we would like to provide a comprehensive perspective of how senescence participates in liver carcinogenesis (Figure 3).

### 2.1. Cellular Senescence in Hepatocyte Injury

Hepatocytes represent the major cell population in the liver. Once hemostasis is perturbed, hepatocytes can sense and respond to detrimental stimulations. When rapid and/or severe, prolonged injuries occur in the liver, the regenerative capacities of the liver may be insufficient. Instead, hepatocytes undergo non-programmed death (e.g., necrosis), programmed death (e.g., apoptosis) and senescence [25]. During severe acute liver injury, senescence is strongly induced in hepatocytes. As a well-known medical compound, excessive paracetamol exposure may affect hepatocytes, which then upregulate senescence-associated β-galactosidase (SA-β-gal), p21 and γH2AX [26]. Similarly, the presence of senescent hepatocytes is regarded as a hallmark of chronic liver diseases, together with sustained liver damage and excessive accumulation of scarring tissue and ductular reaction [27]. In addition, senescent hepatocytes have also been reported in hepatitis B and C virus (HBV and HCV) infection, as well as in genetic hemochromatosis [28,29].

The consequences of hepatocellular senescence differ depending on the phases of liver injury. In the early phase, senescent hepatocytes exert protective liver preservation. Proinflammatory cytokine/chemokines (e.g., CCL2, IL1, IL6 and IL10) and growth factors (e.g., VEGF and TGF-β) are secreted by senescent cells, which further eliminate stimulus and improve tissue healing [30,31]. Nevertheless, when the acute injury turns to chronic, senescent hepatocytes gradually alter the cellular phases of adjacent cells via SASP secretion [32]. Benefits are given to induce cell proliferation in impaired hepatocytes, while healthy hepatocytes maintain a quiescent status. Therefore, under this senescent feedback loop, abnormal proliferation may be activated in hepatocytes with a high risk of carcinogenesis. Moreover, thriving SASP emerges to favor fibrogenesis and inflammation persistence. Manifesting as eventual progressions, cirrhosis and liver function deterioration lead to irreversible malignancy in the liver [33]. Hepatocellular senescence significantly participates in liver disease progression, continuously modelling the microenvironment through SASP, which may eventually cause fibrosis and hepatocellular carcinoma [34]. It has been reported by Wiemann et al., that the fibrotic scarring at the cirrhosis stage is a consequence of hepatocyte telomere shortening and senescence, which also implies that hepatocellular senescence is a promising marker of cirrhosis [35].

### 2.2. Cellular Senescence in Hepatic Fibrosis/Cirrhosis

During liver carcinogenesis, the risks of hepatic fibrosis have been widely studied. Indomitable hepatic fibrosis advances to irreversible hepatic cirrhosis, eventually leading to liver cancers with a latent progression. As the intimate sensor and reactor during fibrogenesis, hepatic stellate cells (HSCs) can be activated and become myofibroblasts (MFBs), and secrete excessive amounts of extracellular matrix (ECM) and matrix metalloproteinases (MMPs) [36]. The fibrosis/cirrhosis progression following the activation of HSCs is prone to be triggered by cytokines/chemokines, which are mainly secreted by activated immune cells (e.g., macrophages and dendritic cells) and abnormally disrupted hepatocytes due to steatosis and senescence [37,38,39].

Among SASP factors, TGF-β and several interleukins have been found as fundamental inducers of HSC activation. TGF-β is responsible for the transdifferentiation of HSCs to MFBs [40,41]. IL-17 directly induced the production of collagen type I of HSCs by activating the signal transducer and activator of the transcription 3 (Stat3) signaling pathway [42]. Fabre et al., found that blocking either IL-22 or IL-17 production in a mouse model resulted in fibrosis amelioration, thus identifying a pathogenic role of IL-17 and IL-22 in driving liver fibrosis [43].

Multiple factors have been identified to be capable of sensitizing HSCs to DNA damage. Krizhanovsky et al., found that the senescence program activated in HSCs limited the fibrogenesis response to acute tissue insult [44]. Intriguingly, Kong et al., reported that IL-22 can protect HSCs from apoptosis by converting cell status into senescence, further exerting an anti-fibrotic effect [45]. Another important SASP factor, IL-10, has also been revealed as an inducer of HSC senescence, which is achieved by IL-10 activating the STAT3-p53 pathway [46], while p53 silencing can block this effect [47]. Consistently, HSCs administrated by alanine-serine-cysteine transporter type-2 (ASCT2), Mannan-binding lectin (MBL) and a PPARγ agonist (15d-PGJ2) resulted in liver fibrosis attenuation through senescence induction in HSCs [48,49,50].

In contrast, other studies reported the detrimental roles that senescence might play in liver fibrosis. Yoshimoto et al., reported that deoxycholic acid (DCA) induced DNA damage in HSCs and promoted HCC progression by SASP-associated tumor-promoting factors [51]. Liu et al., revealed that HCC patients with higher *S. maltophilia* abundance induced the active SASP secretome of HSCs and cirrhosis, provoking the process of hepatocarcinogenesis [52]. As recently released, Yamagishi et al., emphasized that Gasdermin-D-mediated pore formation significantly upregulated IL-1β and IL-33 secretion, which acts as a promotor in non-alcoholic steatohepatitis (NASH)-associated HCC progression [53,54]. In addition, Gasdermin-D was also indicated to regulate macrophage-associated liver fibrosis progression [55].

To sum up, both SASP-induced HSC activation and cellular senescence within HSCs take an active part in hepatocarcinogenesis. Overall, HSC-associated senescence may be regarded as a protector against liver fibrosis and HCC. On the other hand, senescent HSCs can also fuel the tumor environment through multiple SASP factors. Controversial findings address many previous questions, but also bring more.

### 2.3. Cellular Senescence in Hepatic Inflammation

Overall, cellular senescence participates in hepatic inflammation via the SASP secretome. The concept of “senescence surveillance” was proposed to define that hepatocellular senescence induces inflammation to prevent tumor initiation by SASP [56]. The CCL2 chemokine takes an important part in hepatic SASP to recruit circulating monocytes, so that senescent hepatocytes can be efficiently eliminated. Nonetheless, this would not happen if the carcinoma has been transformed from hepatocytes. Initiated as the surveillance and alarmer against hepatocarcinogenesis, senescent hepatocytes end up as a persisting barrier, shielding cancer cells from clearance [57]. Moreover, the robust SASP secretome exhausts immune cells and fuels the angiogenesis, cell proliferation and immune tolerance in HCC [22]. In addition, cancer cell-secreted cytokines resist the maturation of recruited monocytes, thus inhibiting the anti-cancer response driven by NK cells [57,58]. In this setting, senescent hepatocytes behave as a favorable regulator of hepatic homeostasis if they are not in the vicinity of ‘bad’ neighbors such as tumor cells.

Other than circulating monocytes and NK cells, immune cells are broadly involved as either responders or implementers, which subsequently contribute to various outcomes. The telomere shortenings, phenotypic change and cell-cycle arrest in T cells lead to senescence [59,60]. Consequently, T cell cytotoxicity is compromised with the relevant immune responses [61,62]. Notably, high levels of senescent T cells were revealed even in young patients with autoimmune disease and chronic viral infection [63]. In the liver, lymphocyte senescence has been observed in chronic hepatitis, cirrhosis and HCC [64,65,66,67,68]. In addition to functional alterations, it is intriguing to focus on phenotype changes in senescent T cells. Low CD28 expression has been known as an apparent feature. In addition, high expression of Tim-3, CD57, killer cell lectin-like receptor subfamily G member 1 (KLRG-1) are thought as a result of senescence program [69,70,71,72,73]. These findings will assist the development of anti-cancer therapies targeting T-cell senescence.

Till now, the ‘senescent macrophage’ is still a controversial concept. Although senescence markers (e.g., p16^INK4a^ and SA-β-Gal) can be found in macrophages, they do not actually enter senescence processes as assumed [74,75]. In a senescence mouse model with selective Ercc1 depletion, SASP-like phenotypes were suggested in macrophages including evaluated secretion of TNF-a, IL-6 and IL-1β [76]. Nevertheless, liver-resident macrophages (also known as Kupffer cells, KCs) may not only be the earliest immune cell population sensing hepatocellular senescence, but also the frontline of senescence surveillance [77,78]. Bird et al., reported that KCs favor senescence spreading during acetaminophen-induced hepatocyte injury [26]. In addition, the differential phenotype shift was induced in KCs by senescent HSCs and their released SASP factors (e.g., IFN-γ and IL-6), which eventually promoted immune surveillance [79]. 

To summarize, immune cell-associated senescence has been investigated from some aspects. However, how we decipher senescence occurred in immune cells remains obscure, and more studies are expected. 

### 2.4. Cellular Senescence in HCC

In recent decades, senescence has been known as an anti-tumoral mechanism. Given the impediment effect of senescence on tumor development, senescence as an inducible defense mechanism can be exploited to initiate stable proliferation arrest in cancer cells and trigger immunosurveillance to eradicate senescent pre-malignant cells and their neighboring non-senescent pre-malignant cells as a bystander effect during HCC carcinogenesis [34]. For instance, during the process of liver fibrosis, activated HSCs undergo senescence which is reinforced by a self-augmented autocrine loop involving various soluble factors (e.g., TGF-β1, endothelin-1, PAF, angiotensin II, MCP-1, CCL5, IL-10, IL-8) [80]. Senescent activated HSCs in the carbon tetrachloride (CCl4)-induced liver fibrosis exert an anti-fibrosis effect by limiting the excessive proliferation of activated HSCs, reducing extracellular matrix deposition while producing extracellular matrix-degrading enzymes, as well as triggering NK cells to eliminate senescent activated HSCs, all of which contribute to fibrosis alleviation [44,81,82]. Therefore, senescence can be used as an anti-fibrotic therapy to reduce the risk of HCC occurrence. Exogenous administration of adenovirus-expressing IL-22 induces HSCs to enter senescent status, thereby alleviating alcohol-induced fibrosis in mouse liver and accelerating the resolution of liver fibrosis [83]. Doxazosin, an antagonist of the α-1 adrenergic receptor, reverses HSCs activation by inducing their senescence [84]. Whether these senescence-inducing anti-fibrotic reagents reduce the transformation or occurrence rate of HCC in vivo requires further investigation.

Apart from the contribution of senescent activated HSCs to lowering the risk of HCC, *Nras^G12V^*-induced senescent hepatocytes, as pre-malignant cells, can also account for the prevention of HCC occurrence by triggering *RAS*-specific Th1 CD4^+^ T cells to activate macrophages to perform immune clearance of OIS hepatocytes [34]. Senescence immunosurveillance of pre-malignant hepatocytes also requires the recruitment and maturation of CCR2^+^ myeloid cells, which activate macrophages to perform phagocytosis of these pre-malignant hepatocytes to restrict HCC occurrence [57]. CXCL14 from SASP enables a dominant macrophage-based immunosurveillance of *KRAS^G12V^*-induced senescent hepatocytes, acting as a barrier against HCC outgrowth [85]. The preventive role of senescent hepatocytes in HCC occurrence also depends on the duration and severity of liver insults. Acute severe liver injury, rather than chronic moderate liver insult, leads to senescence induction among hepatocytes and activation of macrophages and NK cells, which clear senescent pre-malignant hepatocytes and reduce hepatocarcinogenesis [86].

During carcinogenesis, oncogenes are widely activated, inducing proliferative stress and senescence. Furthermore, oncogene-induced senescence can suppress tumor progression [87]. In the liver, the establishment of senescence in premalignant hepatocytes has been known to limit cancer development. The SASP-dependent recruitment of immune cells eliminate senescent hepatocytes and prevent malignant transformation. Then, the induction of senescence by means of ROC1 ablation can suppress further growth of malignant hepatocytes [34]. Tordella et al., found that SWI/SNF-associated hepatocellular senescence exerted protective effects in the mouse HCC model [88]. Xue et al., revealed that p53 deficiency is necessary for the aggressiveness of HCC, and cellular senescence together with the innate immune system act as a suppressor of tumor growth [89]. Moreover, Zhu et al., uncovered a viral mechanism for HCC that the deregulation of senescence promotes hepatocarcinogenesis [90]. Collectively, the major challenge for potentiating senescence as a safeguard in cancer prevention is to develop senescence inducers that specifically target organs or cell types of interest, which requires further screening of potential chemical or biological candidates in various pre-neoplastic liver diseases.

Recently, it is becoming distinct that senescence may also play a trigger in liver carcinogenesis. Important roles of senescent cells have been determined in the development and progression of cholangiocarcinoma (CCA) via the SASP secretome [91]. Furthermore, it has also been associated with HCC development [92]. Hepatocellular senescence contributes to the incidence of mutations (e.g., p. 53), which can cause carcinogenesis and aggressiveness of HCC [93]. In addition to the importance of instinct senescence escape of *Nras^G12V+^* premalignant hepatocytes in HCC occurrence, immunosurveillance is also an unneglectable factor in the transformation of OIS hepatocytes [94]. Either partial hepatectomy-induced acute injury or thioacetamide-induced chronic liver damage can lead to immunosurveillance failure of OIS hepatocytes, which gives rise to HCC [94]. Karabicici et al., revealed that Doxorubicin-induced senescence promotes stemness and tumorigenicity in an HCC cell line [95]. Huang et al., described that the hepatic SASP induced the activation of macrophages during hepatocarcinogenesis, thereby promoting the progression of HCC [96]. In addition, several long non-coding RNAs (lncRNAs) have been discovered as anti-tumor factors in HCC by regulating cellular senescence. Xiang et al., indicated that PINT87aa (a lncRNA) was significantly increased in the hydrogen peroxide-induced HCC cell senescence model, potentially suppressing HCC progression [97]. Zhao et al., reported that MIAT (a lncRNA) functioned as competitive endogenous RNA (ceRNA) to induce the cellular senescence and tumor-suppressive pathways (e.g., p. 53/p. 21 and p. 16/pRb) [98].

In terms of the SASP and immune surveillance, the effects of senescence may differ with the stages of carcinogenesis. As we have discussed above, SASP and immune surveillance programs can eliminate abnormally proliferating cells, whereas they turn into pro-malignant accomplices with the tumor onset. Accordingly, potential senescence-targeting anti-cancer strategies should be specialized with a precise understanding of cellular senescence in variable phases.

## 3. Cellular Senescence in HCC Therapies

### 3.1. Senescence Induction as an Anti-Cancer Therapy

Senescence incapability due to *Tp53* mutations accelerates HCC onset in a zebrafish model [93]. Likewise, senescence escape renders malignant cells to be more aggressive, which was observed in lymphomas and HCCs [95,99]. On the other hand, senescence restoration by modulating *p53* can retard the proliferation of HCC cells in vitro and the development of HCC in vivo under the SASP-activating immunosurveillance [89,100]. The nature of growth arrest and pro-immunosurveillance secretome of senescent cells have inspired the development of senescence-inducing reagents in malignancy treatment. These reagents, including cell-cycle inhibitors (e.g., CDK4/6 inhibitors: Palbociclib, Abemaciclib and Ribociclib), telomerase inhibitors, cytokines, epigenetic modulators, DNA-damaging chemicals and specific factors that activate the p53/p21-Rb or p16^INK4a^-Rb pathway, have been evaluated in the pre-clinical studies and clinical trials for their therapeutic value in cancer (Table 1) [101].

#### 3.1.1. Cell-Cycle Inhibitors

As approved by the FDA for the treatment of advanced breast cancer, CDK4/6 inhibitors (Palbociclib, Abemaciclib and Ribociclib) functionate their anti-cancer effects by controlling tumor growth, which is not exclusively the result of their senescence induction [102]. Palbociclib (PD-0332991) restrains HCC growth in a genetically engineered mosaic mouse model (*Myc*; *p53*-sgRNA) and extends the survival time of mice with human HCC xenografts, the therapeutic effect of which depends on the normal functionality of *RB1* gene in tumor cells [103]. Ribociclib, a novel orally available CDK4/6 inhibitor, synergized with sorafenib to induce cell death of HCC cells in vitro by inducing cell-cycle arrest at the G1 phase [104]. Another CDK4/6 inhibitor SHR6390 is under clinical investigation for the treatment of advanced HCC [105] (Table 1). Moreover, CDK12 inhibitor THZ531, is also capable of synergizing with sorafenib in HCC treatment by inducing senescence and apoptosis of HCC cell lines [106]. CDK1/2 inhibitor Xylocydine, suppressed the growth of HCC xenografts in BALB/c-nude mice [107]. However, Xylocydine also impeded the function of CDK7/9, which requires a more comprehensive investigation of its role in senescence induction and HCC inhibition.

Cell Division Cycle-7 (CDC7) is a protein kinase that mediates the initiation of DNA replication and G1/S transition in the cell cycle. The anti-depressant sertraline, as an antagonist targeting CDC7, selectively induces senescence of the *TP53*-mutant-bearing HCC cells and promotes the apoptosis of senescent HCC cells when combined with the mTOR inhibitor, leading to reduced HCC growth in the mouse model [108].

AZD7121, an orally bioavailable, vascular endothelial growth factor receptor-2 (VEGFR-2) tyrosine kinase inhibitor can be exploited as a senescence inducer in retarding tumor growth [109] and showed limited toxicity in patients with advanced HCC in a phase II clinical trial [110].

#### 3.1.2. Telomerase Inhibitors

As one of the features of replicative senescence, telomere attrition was found enriched in tumor tissues from patients with non-proliferative HCCs, and was positively associated with various risk factors including ageing, excessive alcohol consumption and liver fibrosis [108]. Somatic mutations in the *TERT* (telomerase reverse transcriptase) promoter were also found to occur in a higher frequency among the HCC lesions with short telomeres compared to those with long telomeres. Moreover, anti-*TERT* antisense oligonucleotides, as a telomerase inhibitor, induced telomere shortening, DNA damage and apoptosis of HCC cell lines, and exhibited therapeutic effects in a xenograft mouse model [111]. Other telomerase inhibitors, such as isothiocyanate, caused cell-cycle arrest and apoptosis of HCC cell lines in vitro, and retarded HCC growth moderately in a nude mouse model [112,113].

#### 3.1.3. Cytokines

Some types of immune cells are capable of inducing senescence in their target cells by secreting various cytokines, so-called cytokine-induced senescence [114,115]. For instance, Th1 lymphocytes secrete IFN-γ and TNF-α to provoke senescence in their recipient β-cancer cells, thereby retarding tumor growth [116]. The senescence-inducing effect was replicable in other types of cancer cells (e.g., melanoma, breast cancer) receiving combined treatment with recombinant IFN-γ and TNF [116]. A follow-up study using tumor antigen-specific T-helper cells in treating β-cancers discovered that intravenous transfer of these Th1 cells exhibiting a better efficiency in senescence induction compared to intraperitoneal administration, which proposed a potential influence of different administration routes of exogenous cytokines on senescence induction in vivo [117]. The important role of IFN-γ and TNF-α in senescence induction has also been verified in another study using adoptive transfer of γδ T cells to treat solid tumor cell lines [118]. Furthermore, other cytokines may also participate in tumor inhibition, which is partly due to their ability to induce senescence. For example, apoptotic lymphoma cells induce M1-polarization of macrophages, which secrete TGF-β to trigger a senescence program in their surrounding lymphoma cells, leading to tumor regression [119]. On the other hand, impediments of senescence-impairing factors may restore the senescence program. For instance, given the senescence ablation effect of CD11b^+^Gr-1^+^ myeloid cells on the *PTEN*-loss-induced senescent prostate cancer cells, an anti-CXC chemokine receptor 2 (CXCR2) antagonist targeting these myeloid cells can sensitize tumor cells to docetaxel-induced senescence, which ultimately leads to a better treatment outcome of prostate cancer in a mouse model [120]. Finally, yet importantly, immune checkpoint blockades can reinforce senescence induction to enhance the immunosurveillance of cancer cells, which has been discussed in detail in these reviews [121,122]. To date, cytokine-induced senescence is understudied in terms of HCC. One interesting study showed that pro-inflammatory factors (TNF-α, IFN-β and IFN-γ) induce senescence of biliary epithelial cells in an ex vivo culture model, which may inspire more exploration regarding cytokine-induced senescence in other cell types during liver diseases [123].

#### 3.1.4. Epigenetic or Genetic Modulators

WM-8014, the inhibitor of histone acetyltransferase KAT6A, potentiates oncogene-induced senescence in vitro without triggering DNA damage and suppresses HCC growth in a zebrafish model [124]. VO-OHpic, a PTEN antagonist, can induce Hep3B cells (with a low baseline expression of PTEN) into senescence and retard the in vivo growth of Hep3B cells in nude mice [125]. The obstacle to using epigenetic or genetic modulators as a senescence-inducing anti-cancer therapy is the low specificity of these chemicals to tumor cells, which might be circumvented by the tumor-orienting biological delivery systems, such as encapsulation with nanoparticles or conjugation with tumor antigen-targeting antibodies.

#### 3.1.5. Other Senescence Inducers

DNA-damaging chemotherapy and radiotherapy also trigger a premature stress-induced senescence program in cancer cells (so-called treatment-induced senescence) [126], which is not the dominant but unneglectable factor that influences treatment response. Increased with age and development of liver cirrhosis and HCC, macroH2A1 was found to synergize with chemo-drug 5-aza-deoxycytidine (5-aza-dC) to induce global DNA hypomethylation while retarding 5-aza-dC-induced senescence, which ultimately ablated the anti-proliferation effect of 5-aza-dC on HCC cells in vitro [127]. This study highlights the impacts of treatment-induced senescence in reflecting the treatment response and the potential of senescence-inducers as treatment sensitizers. Even though senescent cancer cells are less apoptosis prone due to their upregulated expression of anti-apoptosis proteins (e.g., Bcl-W, Bcl-X_L_ and BDNF) [128,129], senescence inducers may alter treatment response of HCC cells to sorafenib, the standard-of-care for advanced HCC. For instance, SHP099, an antagonist targeting Src homology 2 domain-containing phosphatase 2 (SHP2), induces a senescence program in sorafenib-resistant HCC cell lines, leading to vulnerability reacquisition of cancer cells to sorafenib in HCC xenografts and survival extension in a sorafenib-refractory immune-competent mouse model with Nras/Akt-induced HCC when being combined with Sorafenib [130].

Collectively, senescence induction as a regimen for cancer treatment shed light on decelerating tumor progression and extending survival time. The potential side effects of senescence induction in cancers might be local inflammation in a short-term period or systemic inflammation due to the persistence of senescent cells, which leads to physiological function deterioration of organs that are susceptible to senescence inducers, resulting in bone marrow suppression, cardiac dysfunction, etc. [131]. Moreover, attention needs to be paid to the agents and methods for inducing senescence as a cancer prevention or treatment strategy. For instance, HSCs turned senescent upon receiving deoxycholic acid and lipoteichoic acid (two metabolites from obesity-associated Gram-positive bacteria). As a result, these senescent HSCs initiated an accelerating process of hepatocarcinogenesis [51,132]. Moreover, senescence-associated reprogramming renders cancer cells to behave more aggressively once escaping from their cell-cycle arrest [99]. In the immunocompromised mouse model bearing HCC xenografts, chemotherapy-induced senescence enables the non-stem cancer cell population (EpCAM^−^/CD133^−^) to acquire stemness and tumorigenicity features in vivo through SASP factors [95]. Therefore, a comprehensive investigation of individual senescence components in cancer development is required to fully understand and manipulate the spatial-temporal-contextual role of senescence in HCC.

### 3.2. Elimination of Detrimental Senescence as an Anti-Cancer Therapy

Given the aforementioned bi-directional roles of senescent cells and their secretory phenotypes on tumor prevention and promotion, eradicating detrimental senescent cells after chemotherapy (so-called ‘Combo strategies’, Figure 1) or manipulating SASP towards an anti-tumor direction may improve treatment outcomes in cancers [101]. Pre-clinical and clinical studies are ongoing using these strategies (Table 1).

#### 3.2.1. Senolytics

Direct killing of detrimental senescent cells using pharmacological intervention, namely senolytics, has shown to be an effective anti-HCC strategy [133]. The major category of senolytics targeting the inactive apoptosis pathways in senescent cells has been shown to be potent in improving cancer control, especially in the sequential application of senescence-inducing cancer therapy followed by senolytic regimens [134]. These senolytics include the Bcl2 family antagonist ABT-263 (Navitoclax) and ABT737, multi-kinase antagonist Dasatinib, p53-regulated apoptosis inducers FOXO4-DRI and UBX0101 (MDM2 antagonist), as well as PI3K-mediated survival inhibitors, Quercetin in combination with Dasatinib (D + Q), and Fisetin [135]. The regimen of D + Q or ABT-263 alleviates tumor development by selectively inducing apoptosis of senescent HSCs that contributes to HCC growth in mice with hepatocyte-specific loss of the *Fbp1* gene [136]. Furthermore, the combination of D + Q can clear senescent hepatocytes to alleviate hepatic steatosis [137], which might be a potential therapeutic reagent to eliminate senescent hepatocytes that promote HCC progression by preventing maturation of the recruited myeloid precursors [57]. Indeed, D + Q-treated aged mice present a lower risk of HCC [138]. However, it was reported that D + Q failed to enhance the efficacy of senescence-inducing chemotherapy (doxorubicin) in an immunocompromised mouse model with liver cancer [139]. Even so, conclusions should not be drawn immediately for the failure of this regimen in HCC treatment, since not only the intrinsic apoptosis program but also the senescence-triggered immune response play a critical role in senescence elimination. We must keep in mind that senolytics may function more properly in the existence of intact immunity. Furthermore, some reagents might be utilized as senolytics sensitizers to treat HCC after arginine deprivation-induced senescence [140]. Another category of senolytics is to exploit the aberrant accumulation of β-galactosidase in senescent cells. For instance, the β-galactosidase-targeting prodrug SSK1 can be metabolized by β-galactosidase that massively accumulates in senescent cells, leading to cell death, which eradicates senescent liver cells and attenuates hepatic fibrosis [141]. The preventive effects of SSK1 in HCC requires further investigation.

Attention should be paid to the weak specificity of senolytics towards deleterious senescent cells, which may cause extensive removal of senescent cells in vivo, leading to undesired serious side effects (e.g., thrombocytopenia and lymphopenia) in clinical trials [133,142]. Therefore, the precise elimination of unwanted senescent cells in vivo needs to be further investigated in HCC, or using an alternative strategy such as senomorphics, to modulate specifically one of the components or regulators of SASP.

#### 3.2.2. Senomorphics

Senomorphics are a group of compounds that remodel the senescence microenvironment by altering SASP without disturbing the integrity of senescent cells and their anti-tumor features [143]. One strategy to modify or reduce SASP production is to inhibit the SASP release machinery Gasdermin D [54] or manipulate the critical regulators of SASP, such as mTOR (rapamycin) [144,145], p38-MAPK (BIRB-796 and UR-13756) [146], NF-κB (BAY 11-7082) [147], JAK2/STAT3 (Ruxolitinib) [148] and other small molecules that target BRD4, L1 and STING [19].

The mTOR signal pathway is active in senescent cells, which can be targeted to unleash potential cellular vulnerabilities of these senescent cells towards lethal stimuli [144]. For example, sertraline-induced senescent HCC cells with TP53-mutants can be eliminated by the mTOR inhibitor [108]. The combination of sertraline and mTOR inhibitor yields a better prognosis of HCC-bearing mice compared to the standard-of-care Sorafenib [108]. Another example is using inhibitors to target cell-cycle checkpoints that mediate chromosome separation during mitosis, such as spindle assembly checkpoint (SAC), thereby inducing DNA damage and senescence-like switch in dividing cells. An orally active small molecule, CFI-402257, that antagonizes threonine tyrosine kinase (TTK), which is a critical regulator of SAC, was found effective in retarding HCC growth in vivo. The underlying mechanism of CFI-402257 as an HCC suppressor was achieved by its induced SASP through the DDX41-STING cytosolic DNA sensing pathway. The CFI-402257-induced SASP further attracted various subsets of immune cells (NK cells, CD4+ T cells, and CD8+ T cells) to control HCC [149].

However, the therapeutic value of these signal regulators is ambivalent due to the controversial properties of SASP in influencing cancer development at different disease stages. Hence, targeting specific components of SASP, such as IL-6 or IL-8, serves as another strategy to reverse the tumor-propagating or enhance the tumor-suppressing function of SASP. PGE2 secreted by senescent HSCs binds to its receptor PTGER4 on immune cells to suppress CD103+ dendritic cells and reduce activation of CD8+ T cells, thereby contributing to the progress of HFD-induced HCC due to the immunosurveillance failure [132]. Therefore, the anti-PTGER4 inhibitor AAT-008 efficiently attenuates HCC development in a mouse model by restoring the anti-cancer immunosurveillance function of DCs and cytotoxic T cells [132]. Despite that these SASP components are not exclusively expressed or secreted by senescent cells, the targeted regulation of these molecules may indeed yield promising outcomes following the senescence-inducing anti-cancer therapies, especially after the precise spatial-temporal description of SASP in different tumor entities.

#### 3.2.3. Senescence-Targeted Immunotherapy

In addition to SASP, senescent cells also present a unique surface proteome that involves in their interaction with other cell types including immune cells. Scott W. Lowe’s group found that senescent liver cancer cells upregulated the receptors of type II interferons (IFN-gamma) to become hypersensitive to IFN-gamma, which activated its downstream signals and enhanced the antigen process and presentation on MHC-I molecules, leading to a more “visible” feature of these senescent liver cells to immunosurveillance in vivo [150]. Therefore, senescence-inducing therapy combined with senescence-targeting immunotherapy is also a promising way to overcome senescence-associated immunosuppression and hence eradicate immuno-escaping malignant cells [19]. An increasing number of studies have shown the altered expression of senescence-associated surface proteins in senescent cells, including some immunoregulatory factors, which inspires the investigation of novel immunotherapies to remodel the permissive immune microenvironment and eliminate senescent cells. For instance, immune checkpoint proteins, a group of senescence-associated proteins, can be targeted to eliminate senescent malignant cells [151]. Given the destabilization effects of cyclin D-CDK4 kinase on PD-L1 protein [152], therapeutic application of the PD-L1/PD-1 blockade has been verified in the CDK4/6 inhibitor-induced senescence scenarios in various malignancies including colorectal cancer [152], melanoma [153,154] and lung cancer [155]. Furthermore, anti-PD-1 neutralizing antibodies can sensitize senescent cells, especially PD-L1+ senescent cells, to CD8+ T-cell surveillance, and ameliorate the pathogenesis of high-fat-diet-induced non-alcoholic steatohepatitis (NASH) in the mouse model [156]. The therapeutic value of PD-L1/PD-1 blockade in HCC with enhanced senescence signature after sorafenib needs to be addressed, together with the immune landscape alteration in HCC lesions.

Moreover, HLA-E, a surface protein found to be upregulated in liver tumors and also on cell surfaces of senescent cells, suppresses the effector function of NK cells and CD8+ T cells by binding to its receptor NKG2A on these immune cells, which contributes to the in vivo accumulation of senescent cells [157]. Anti-NKG2A neutralizing antibody (Monalizumab) has been demonstrated to boost NK cells to lyse malignant cells in combination with EGFR-inhibitor (Cetuximab) in a phase II trial, and further restore the cytotoxicity of CD8+ T cells when in combination with anti-PD-L1 blocking antibody (Durvalumab) [158].

Studies have shown that MICA and MICB (stress-induced proteins as NKG2D ligands) are upregulated in hepatocytes from chronic liver diseases and tumor cells from HCC lesions [159]. It was reported that MICA and MICB can be proteolytically cleaved in senescent cells and hence allowed senescent cells to escape NK cell/macrophage-mediated immunosurveillance [160], which can be abolished by the antibody that targets the α3 domains of MICA/MICB and prevents their shedding, leading to tumor regression in the mouse models of melanoma and acute myeloid leukemia [161,162]. Unfortunately, neither the anti-NKG2A nor anti-MICA/MICB neutralizing antibodies have been evaluated in anti-HCC senescence-inducing treatment scenarios, which urges researchers to investigate their therapeutic roles in HCC.

Apart from immunoregulatory molecules, the specific senescence-associated proteins on cellular membranes can be targeted by antibody- or immune-cell-based immunotherapies, as an alternative approach to restoring functional immunosurveillance of senescent cells. For example, dipeptidyl peptidase 4 (DPP4), the functional form of CD26 (a multifunctional transmembrane glycoprotein), was reported to be expressed exclusively on senescent fibroblasts [163], and act as a pro-inflammatory factor in the liver [164]. The specific anti-DPP4 antagonists can reduce the proliferation and clonogenicity of HCC cells in vitro [165], and impede the growth of HCC xenografts in the immunocompromised mouse. The tumor suppression effect of anti-DPP4 antagonists depended on the immunosurveillance of NK cells [166]. In a retrospective clinical study, anti-DPP4 inhibitors were found to decrease the risk of HCC in patients with chronic hepatitis C infection and type 2 diabetes mellitus (T2DM) [167]. However, another report from the same region suggested that usage of DPP-4 inhibitors was associated with higher risks of decompensated cirrhosis and hepatic failure among patients with T2DM and compensated liver cirrhosis [168]. Therefore, a comprehensive evaluation of disease conditions needs to be taken into account regarding the usage of DPP4 inhibitors in patients with liver dysfunction. Despite that anti-DPP4 antibody can selectively induce NK cell-mediated antibody-dependent cytotoxicity of senescent human diploid fibroblasts [163], whether the anti-HCC effects of anti-DPP4 inhibitors is attributed to their senescence orientation require further investigation.

Another senescence-specific surface protein urokinase-type plasminogen activator receptor (uPAR) was discovered by data mining of the RNA sequence from three independent senescence models (replication-/oncogene-/therapy-induced senescence). Based on this finding, uPAR-specific CAR-T cells were generated and reported to effectively eliminate senescent hepatocytes in the benign disease as well as the malignancy models, leading to the alleviation of liver fibrosis in mice with NAFLD [169]. The promising achievement of this novel CAR-T cell therapy encourages scientists to dissect the surfaceome of senescent cells, especially focusing on the senescence-associated or -specific peptides presented by MHC molecules on the cell surfaces. Correspondingly, the identification of senescence-specific T-cell clones and their TCR sequences may foster the development of immunotherapy to eliminate undesirable senescent cells [170].

Abbreviations: OS: overall survival; ORR: Objective Response Rate; PFS: Progression-Free Survival; wk: week; mon: month.

## 4. Discussion and Perspectives

Cellular senescence has been determined to broadly participate in a wide variety of mammal cell processes. The SASP secretome allows senescent cells to function as key influencers of their microenvironment, which can vigorously address the question we proposed in the title: senescent cells are not just passengers, but seem to play active, driving roles in disease progression. Other than well-known approaches (cytokine-induced senescence, treatment-induced senescence, oncogene-induced senescence and replication-induced senescence), virus-induced senescence and metabolite-induced senescence have set off a new tide recently [87]. Deoxycholic acid (DCA), a gut bacterial metabolite known to cause DNA damage, has been reported to provoke SASP in HSCs and tumor development [51]. Caloric restriction can shape the gut microbiome and delay immune senescence [171]. These two findings reveal promising roles of gut microbiota in senescence aspects of liver and immune reaction. In addition, hepatitis viruses have been implicated in both carcinogenesis and senescence [29,172]. The hepatitis B virus (HBV) x protein (HBx) can both induce cellular senescence and promote SASP phenotype in hepatic cells [173,174,175,176]. Cellular senescence is associated with the progression of liver fibrosis in chronic hepatitis C patients [177]. Hepatitis C virus (HCV) is also determined to induce T cell senescence and oxidative stress in the liver, which contribute to hepatocarcinogenesis [177,178].

Immune-checkpoint blockade atezolizumab, a humanized anti-PD-L1 IgG1 antibody, combining with anti-VEGF-A antibody bevacizumab, has been approved by the FDA for the treatment of advanced unresectable or metastatic HCC [179,180]. Moreover, anti-PD-1 antibodies nivolumab and pembrolizumab were also approved for the treatment of advanced HCC as the second-line regimens when combined with the anti-CTLA4 antibody ipilimumab [181] and administrated as a monotherapy [182], respectively. Other immune-checkpoint blockades including anti-PD-L1 antibody (durvalumab) and anti-cytotoxic T-lymphocyte-associated protein 4 (CTLA4) antibody (tremelimumab) were under clinical investigation as a combined therapy and revealed to improve the progression-free survival of patients with unresectable HCC [183]. However, the primary resistance to these immune-checkpoint blockades is the major issue influencing clinical decision-making. Though the combination of immune-checkpoint blockades and anti-angiogenic agents might ameliorate the unresponsiveness of solid HCC entities towards immunotherapy, efficient predictive biomarkers and novel treatment modalities, such as senescence-associated therapies, are still required to reduce the failure of immunotherapies in HCC [184].

In liver diseases, especially HCC, it is becoming intriguing to uncover the multi-functions of cellular senescence in phase-defined disease progressions. Nonetheless, considering the continuous hepatic injury, fibrosis/cirrhosis and inflammation initiate hepatocarcinogenesis, HCC virtually occurs far earlier than the clinical diagnosis. Thus, comprehensive understanding should be given to those occult precancerous bioprocesses. Moreover, attention should also be drawn to the role of cellular senescence in the drug response of HCC. For instance, the specific antagonist SHP099 targeting Src homology 2 domain-containing phosphatase 2 (SHP2) can reduce sorafenib resistance in HCC cell lines and organoids, the mechanism of which was achieved by the induced senescence [130]. However, more studies are needed to investigate the role of senescence in the drug response of HCC. Further, an increasing number of therapeutic strategies have been developed targeting cellular senescence by selectively triggering or eliminating senescent cells and manipulating SASP. The challenges for utilizing senescence as a weapon to target HCC are the lack of specific senescence markers and understanding the dynamic role of senescence in different disease stages and in different disease contexts. From further perspectives, we expect more senescence-associated diagnostic and therapeutic approaches in precancerous and early stages of HCC.

## Figures and Tables

**Figure 1 cells-12-00132-f001:**
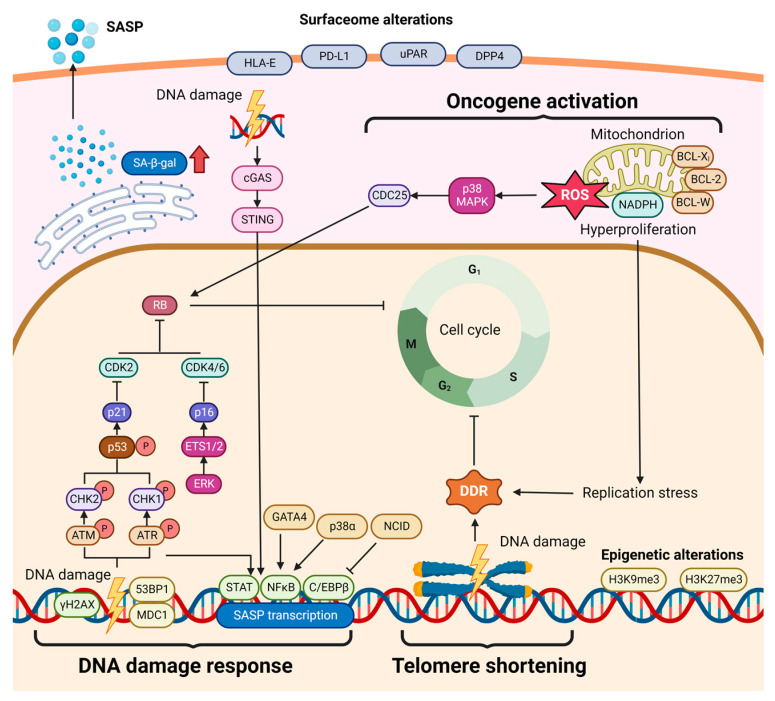
Pathways in senescence induction. DNA-damaging irradiation and chemotherapy, oncogene activation, tumor-suppressor inactivation, oxidative stress and telomere shortening cause DNA damage and activate the DNA damage response (DDR) pathway, which ultimately triggers cellular senescence. DNA double-strain or single-strain breaks activate ataxia telangiectasia-mutated (ATM)/ATM and Rad3-related (ATR) signaling, leading to p53 and p21^CIP1^ activation. In parallel, p16INK4a can also be activated by the ERK/ETS1/2 pathway. Consequently, p21^CIP1^ inhibiting cyclin-dependent kinase 2 (CDK2) and p16^INK4a^ inhibiting CDK4/6 derepress the retinoblastoma (Rb) tumor suppressor, which ultimately retards cells to enter S phase from G1 phase, causing cell-cycle arrest. Moreover, oncogene activation can cause the accumulation of Rb via the p38/mitogen-activated protein kinase (MAPK)/cell division cycle 25 (CDC25) pathway that is activated by massive reactive oxygen species (ROS) in mitochondria. Simultaneously, unsolvable DDR activates the Rb and p53 pathways, and promotes the formation of promyelocytic leukemia nuclear bodies, which ultimately leads to senescence-associated heterochromatin foci (SAHF) through the ASF1A and HIRA chaperones. Furthermore, cytosolic DNA triggers the cyclic GMP-AMP synthase (cGAS)/signaling effector stimulator of interferon genes (STING) DNA sensing pathway, together with the activated DDR pathway, activates several transcription factors including STAT (signal transducer and activator of transcription), NF-kB (nuclear factor ‘kappa-light-chain-enhancer’ of activated B-cells) and C/EBPβ (CCAAT/enhancer binding protein β). These transcription factors regulate a series of target genes coding various senescence-associated proteins, which are secreted to the ultra-space of senescent cells, known as senescence-associated secretory phenotype (SASP). Overall, these senescence-related pathways are interdependent and form a complex regulatory network, contributing to several senescence phenotypes, including p21 and p16 upregulation, stable cell-cycle arrest, β-galactosidase (β-gal) accumulation, SASP production, apoptosis resistance (upregulation of anti-apoptosis proteins: Bcl-X_L_, Bcl-2 and Bcl-W), epigenetic alteration (tri-methylation at the 9th or 27th lysine residue of the histone H3 protein: H3K9me3 and H3K27me3) and surfaceome changes (upregulation of HLA-E, PD-L1, uPAR, and DPP4).

**Figure 2 cells-12-00132-f002:**
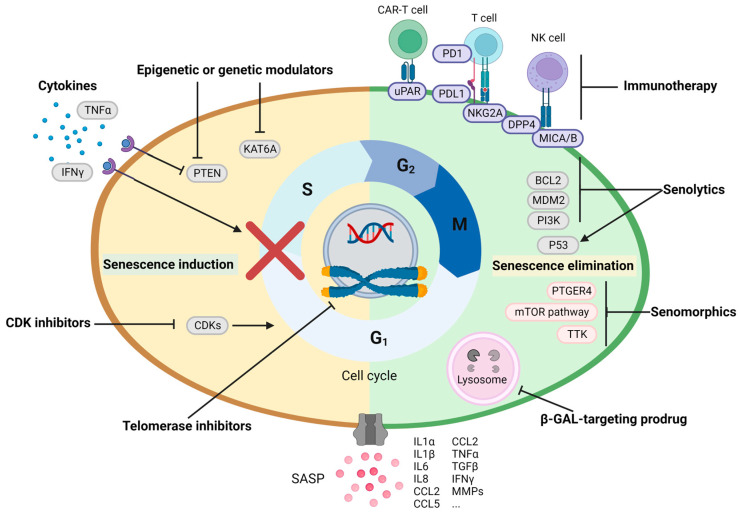
‘Combo strikes’ targeting senescence as an anti-cancer strategy. Treating cancer with chemotherapy, irradiation or cell-cycle inhibitors to induce senescence (pro-apoptosis may be the major anti-cancer effect of chemotherapy and irradiation), which is accompanied by the immunosurveillance of these senescent cells. However, some senescent cells escape from being eliminated by the immune cells (NK cells, macrophages and cytotoxic T cells). The surviving senescent cells may cause chronic inflammation, thereby creating an immunosuppressive microenvironment to allow cancer relapse or self-reprogramming to gain stemness signature and senescence escape. To further prevent disease relapse, these detrimental senescent cells can be removed by the pro-apoptosis or anti-survival senolytics, β-galactosidase-targeting prodrugs, SASP-regulating senomorphics, as well as antibody-based and immune cells-based immunotherapy.

**Figure 3 cells-12-00132-f003:**
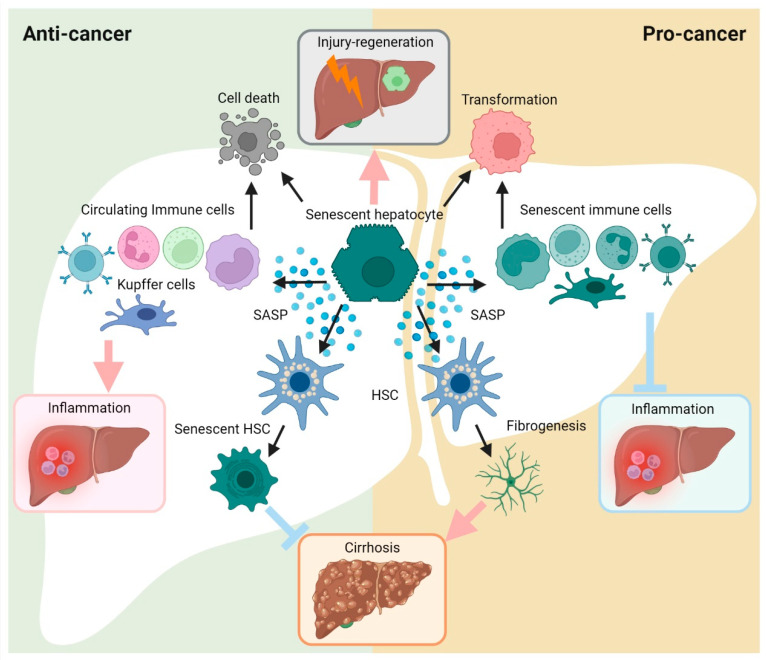
Diverse roles for cellular senescence in hepatocarcinogenesis. The cellular senescence triggered in hepatic cells can arrest cell-cycle progression, subsequently leading to multiple phenotypical alterations. On the one hand, senescent cells can undergo cell death or initiate more active cell proliferation (cancer transformation). On the other hand, the exuberant SASP contributes to modulation of the inflammation and fibrogenesis/cirrhosis. Ultimately, cellular senescence plays as a ‘double agent’ in hepatocarcinogenesis.

**Table 1 cells-12-00132-t001:** Senescence-targeting therapies in clinical trials.

Senescence Inducer						
Category	Compound	Regimens	Phase	Population	Results	Study Number
CDK4/6 Inhibitor	Palbociclib (PD0332991)	Singular treatment	II	23 patients with advanced HCC	OS: 40 (24–96) wk, 11/23 (47.83%) with severe side effects	NCT01356628
	Abemaciclib	Combined with nivolumab	II	7 patients with inoperable HCC	1/7 (14.3% ORR), PFS: 2.1 (1.6 to 5.6) mon, OS: 7.9 (2.6 to 22.7) mon, 3/7 (42.86%) severe side effects	NCT03781960
	Ribociclib (LEE011)	Combined with chemoembolization	Ib/II	5 patients with advanced HCC	1/5 (20.00%) severe side effects	NCT02524119
	SHR6390	Combined with anti-PD-1 Inhibitor SHR-1210	Ib/II	41 advanced colorectal cancer, non-small-cell lung cancer and HCC	Ongoing	NCT03601598
VEGFR-2 tyrosine kinase inhibitor	AZD2171 (Cediranib maleate)	Singular treatment	II	17 patients with locally advanced unresectable or metastatic HCC	OS: 11.7 (7.5 to 13.6) mon, 5/17 (29.41%) severe side effects	NCT00427973
**Senescence eliminator**						
Senolytics	Navitoclax	Combined with Sorafenib (Nexavar)	I	44 patients with relapsed or refractory solid organ tumors including HCC	Ongoing	NCT02143401
	Dasatinib (BMS-354825)	Singular treatment	I	80 patients with unresectable or metastatic solid tumors including HCC	Completed	NCT00608361
	Dasatinib (BMS-354825)	Singular treatment	II	25 patients with unresectable advanced HCC	PFS: 3.7 (1.8 to 6.4) mon, OS: 7.5 (2.9 to 13.6) mon, 13/25 (52.00%) serious side effects	NCT00459108
mTOR inhibitor	Temsirolimus	Combined with Sorafenib	I/II	Advanced HCC	Cancelled	NCT01335074
		Combined with pegylated liposomal doxorubicin (Doxil^®^)	II	Advanced HCC	Cancelled	NCT01281943
	ABI-009	Combined with anti-PD1 inhibitor nivolumab	I/II	26 patients with advanced sarcoma and certain cancers including HCC (with genetic mutations sensitive to mTOR inhibitors)	4/26 (15%) serious side effects	NCT03190174
Anti-DPP4 inhibitor	Sitagliptin	Singular treatment before HCC resection	I	14 HCC patients undergoing liver resection	No severe adverse event	NCT02650427

## Data Availability

Not applicable.

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
