# Peer review of "Cellular Senescence in Hepatocellular Carcinoma: The Passenger or the Driver?"

_cells, 2022, doi:10.3390/cells12010132_

Round 1
Reviewer 1 Report
Cellular senescence is a stable, terminal cell cycle halt condition accompanied by multiple macromolecular alterations and a pro-inflammatory, hypersecretory phenotype. Senescent cell entry can operate as a barrier to carcinogenesis, making it possible for any anticancer therapy to aim toward this result. Contrary to popular belief, research during the last ten years have shown that cancerous and non-cancerous cells can develop pro-tumorigenic traits under specific circumstances and situations. In this Review, the authors provide an overview of the key translational and newly emerging clinical findings before discussing the key mechanisms underlying the antitumorigenic functions of senescent cells. They then consider the cell-intrinsic and cell-extrinsic factors that contribute to their switch towards an HCC-promoting role. Finally, the authors thoroughly explain several senolytic and senomorphic therapies and how they may help liver cancer patients.
Comments
1- Metabolites produced by the gut microbiota may affect a number of growth factors and even promote tumor development by causing cellular senescence. The authors of this review need to discuss the connection between chronic liver disease and the gut microbiota's immune response as well as the impact of the gut microbiota on liver illnesses.
2-Please describe cellular senescence and its role in drug resistance.
3-The Authors also need to illustrate the mechanism or process of cellular senescence in the figure.
4-Role cellular immune cell recruitment in tumor cells and microenvironment should be discussed.
5- The connection between cellular senescence and the development of HCC appears to be made by the influence of HBV and HCV. The landscape for the assessment of chronic liver disease may be improved with the identification and validation of hepatocyte senescence markers, therefore it will be crucial to provide useful information on HBV or HCV-related cellular senescence.
Author Response
To reviewer 1
Many thanks to your comments! We appreciate them and respond below in detail:
- We have included contents in the discussion, regarding comment 1, 2 and 5.
-The conceptional scheme of cellular senescence has been described in new added Figure 1.
- It was initially included in ‘2.3. Cellular senescence in hepatic inflammation’. We described senescence induced inflammation before and after HCC onset.
- English writing of this manuscript has been checked and improved by consulting to a native speaker.
Best regards,
Dr. MD. Hangyang Liu, on behalf of all authors
Reviewer 2 Report
Advantage:
Generally, the present manuscript is with substantial content and significance for senescence and HCC, the authorreviewed the relevant and latest articles about the cellular senescence in hepatocarcinogenesis and the therapeutic medications and strategies for HCC while describing the details of these researches based on drug’s categories. Furthermore, the author summarized the relevant papers and extracted the core data to present a clear outline of the senescence-targeting therapies in clinical trials which should be highly appreciated. This manuscript is well organized and performed in logic. This article presents a very clear and abundant panorama for the understanding of senescence in pathogenesis of HCC and revealed the potential application of targeting senescence in HCC treatment. This provides very valuable information for clinicians and scientists.
Weaknesses:
· A chapter to introduce senescence and drug resistance including chemotherapy, targeted therapy (TKI), and others would complete and enrich this overview.
· Some minor corrections need to be clarified. For example, DDP4 on senescent HCC cells is used as a target to be antagonized by specific neutralizing antibodies for inhibiting HCC growth (Line 520-539). While in Figure 1, DPP4 is not clearly depicted on the senescent cell as it is stated in the related text. Another example is CDKs (cyclin-dependent kinases) in Figure 1. Inhibitors targeting CDKs are used for inducing cellular senescence (Section 3.1.1), which should be illustrated clearly in Figure 1 as a strategy for senescence induction, but not included in the category of “senolytics. Also, Please pay attention to some details such as the consistency of the layout: 3.1.3./3.2.1../3.2.2.. the three chapters are different from others, please check it again.
· Some latest research papers can be included and discussed in this review. For example, Wuestefeld’s work (PMID: 36449018) can be the latest evidence to back up the role of senescence in mediating the malignant transformation of senescent hepatocytes. Moreover, Scott W. Lowe’s work (PMID: 36302222) can further verify the importance of senescence in influencing the anti-tumor immune response in an in vivo liver cancer model. For the senescence-targeting treatments in HCC (Section 3.2.1), some well-conducted research might be note-worthy to be discussed as well (PMID: 35869934, 35839757). Other senomorphics like inhibitors targeting gasdermin D (GSDMD) to reduce SASP secretion can also be mentioned in the review (PMID: 35749514).
· In the Discussion section, the authors can indicate some crucial challenges, such as how to understand the contradictory roles of senescence in hepatocellular carcinoma. This may inspire more interesting thinking for readers.
· The authors should underscore the failure of immunotherapies in HCC as compared to other tumor entities to emphasize the need for new therapeutic targets.
· This review paper is generally well-written. However, minor improvements and corrections need to be made on both grammar and spelling.
Author Response
To reviewer 2
Many thanks to your comments! We appreciate them and respond below in detail:
- We have included contents in the discussion, regarding the comment 1, 4 and 5.
- Revisions have been made accordingly.
- Latest citations have been updated accordingly.
- English writing of this manuscript has been checked and improved, by consulting to a native speaker.
Best regards,
Dr. MD. Hanyang Liu, on behalf of all authors
Round 2
Reviewer 1 Report
I am happy with the Author's responses and current form of the manuscript.